# FERN: A Fetal Echocardiography Registration Network for 2D-to-3D Alignment

**Paula Ramirez**                                    PAULA.RAMIREZ_GILLILAND@KCL.AC.UK
**David Lloyd**                                              DAVID.LLOYD@KCL.AC.UK
**Jacqueline Matthew**                              JACQUELINE.MATTHEW@KCL.AC.UK
**Reza Razavi**                                              REZA.RAZAVI@KCL.AC.UK
**Milou van Poppel**                                  MILOU.VAN_POPPEL@KCL.AC.UK
**Andrew King**                                           ANDREW.KING@KCL.AC.UK
**Maria Deprez**                                          MARIA.DEPREZ@KCL.AC.UK
*School of Biomedical Engineering and Imaging Sciences, King's College London, London, UK.*

**Editors:** Accepted for publication at MIDL 2025

## Abstract

2D Freehand echocardiography remains the primary imaging modality for routine fetal cardiac care, essential in the antenatal detection of Congenital Heart Disease (CHD). However, there is a lack of spatial context which requires 3D imaging. Current 3D methods, such as Spatio-Temporal Image Correlation (STIC), face limitations in success rate, image quality, and ease of use, and come at the cost of lower spatial and temporal resolution compared to 2D acquisitions. This work studies the feasibility of aligning real high spatial and temporal resolution 2D fetal echocardiography into a reference 3D space defined by lower resolution 3D STIC. FERN, a **F**etal **E**chocardiography **R**egistration **N**etwork, employs transformers for standard fetal echocardiography view alignment. The network is trained on simulated 2D slices derived from 3D volumes at end-diastole, and validated on real 2D acquisitions from fetuses with Coarctation of the Aorta and Right Aortic Arch diagnoses, achieving a mean Euclidean distance of $2.98 \pm 1.27$ mm on cardiac region-of-interest points between predicted and manually selected planes. Compared to manually aligned planes, improved image similarity to an average atlas is achieved, confirmed by blinded best plane selection. This work demonstrates that high spatial and temporal resolution 2D fetal echocardiography can be integrated into a 3D context provided by lower-resolution 3D acquisitions or fetal cardiac atlases, potentially resulting in a new 3D visualisation tool for enhanced CHD diagnosis. Code will be available at: https://github.com/PaulaRamirezGilliland/FERN

**Keywords:** Fetal echocardiography, 3D Fetal ultrasound, STIC, Congenital Heart Disease, 3D plane localisation, Fetal cardiology, Fetal cardiac imaging

## 1. Introduction

Congenital Heart Disease (CHD) is the most common group of malformations in fetuses and infants, with an incidence of around 6 per 1000 live births for moderate and severe forms (Hoffman and Kaplan, 2002). Antenatal diagnosis highly impacts patient prognosis, reducing mortality and morbidity (Tworetzky et al., 2000; Bonnet et al., 1999).

2D fetal echocardiography, the primary modality for antenatal CHD detection, offers high spatial and temporal resolution at a low cost. However, it lacks 3D context, requiring sonographers to mentally reconstruct 3D anatomy. Spatio-Temporal Image Correlation (STIC) (DeVore et al., 2003) offers 3D+time ultrasound for fetal cardiac evaluation via

automatic volume sweeps reordered by the cardiac cycle. However, despite over 20 years of availability, it is underused due to significant technological limitations.

STIC is challenging to acquire, with a limited success rate (Inamura et al., 2020) which may lengthen examination times. STIC is anisotropic, with lower spatial and temporal resolution than 2D acquisitions and susceptibility to motion-induced synchronisation errors.

2D Fetal echocardiography acquisitions cover different cross-sectional levels of the cardiac anatomy using a standardised protocol (Carvalho et al., 2023) at an approximately axial position, avoiding shadowing from the fetal ribs. The standard views encompassing the cardiac anatomy include: 4 Chamber view (4CH), Left Ventricle Outflow Tract (LVOT), Right Ventricle Outflow Tract (RVOT), 3 Vessel/3 Vessel and Trachea Views (3VV/3VT).

Aligning standard views into a 3D space is clinically valuable for assessing view quality and distinguishing between anatomical variability, pathological features, probe and fetal motion. This is particularly relevant given the prevalence of the standard view acquisition protocol and the high anatomical variability in CHD, and may be achieved by comparing the estimated plane locations with the corresponding location in an average 3D atlas. This work endeavours to align 2D standard views into an average 3D space using deep learning, to provide 3D context to these high-resolution acquisitions during clinical examinations.

## 1.1. Related Works

Hou et al. (2018) introduced a CNN-based method for 2D to 3D fetal MRI slice localisation into a canonical atlas space, forming the foundation for later approaches using anchor points, point loss, and Fibonacci sphere sampling (Xu et al., 2022; Yeung et al., 2021). Alternative works explored geodesic loss functions (Mohseni Salehi et al., 2019). The most relevant prior work in the literature is PlaneInVol (Yeung et al., 2021), addressing the 3D localisation of 2D Ultrasound brain planes. PlaneInVol utilises a CNN architecture, computing attention across input slices to predict slice transformations. It is trained on dense inputs, i.e. for each case, the spatial context of multiple slices (N=32) is used to generate the prediction and learn the transformations. For this reason, it is sensitive to low numbers of input slices ($< 10$ slices). Unlike a transformer architecture, where positional tokens are an inherent part of the structure, PlaneInVol does not leverage any prior positional information, such as the view type. QAERTS (Ramesh et al., 2024) builds on PlaneInVol, enhancing accuracy and efficiency via a multi-head architecture, where different geometric transformations and their associated variances are learnt. Inputs with low variances are given higher weight.

Slice-to-Volume Registration Transformer (SVoRT) (Xu et al., 2022) is a transformer-based method for reconstructing fetal brain MRI volumes from stacks of slices using an iterative approach, proceeding from classical methods (Kuklisova-Murgasova et al., 2012). In SVoRT, the slices within a given stack are assumed to have highly correlated positions, due to the acquisition protocols of fetal MRI. This work builds on SVoRT and PlaneInVol, adapting the framework for sparse fetal echocardiography data. Such approaches have not been developed for fetal echocardiography data.

## 1.2. Contributions

This study presents and validates FERN, a **F**etal **E**chocardiography **R**egistration **N**etwork for 2D-to-3D alignment, see Fig. 1, combining the spatial context of 3D imaging with the

superior resolution and accessibility of 2D echocardiography. FERN is validated on real CHD cases, laying the foundation for a novel 3D fetal cardiac visualisation tool.

Two novel technical contributions are introduced compared to prior works: (1) the use of a view positional indicator and a transformer architecture for ultrasound plane localization, contrasting with the CNN from Yeung et al. (2021), and (2) the ability to handle sparse inputs consisting of only 1-5 standard view slices in any orientation. In contrast, SVoRT (Xu et al., 2022) requires dense inputs comprising multiple highly correlated slice stacks. Dense inputs means that multiple slices, densely sampling the anatomy of each case, are needed to generate the transformation predictions, rather than 1-5 independent slices.

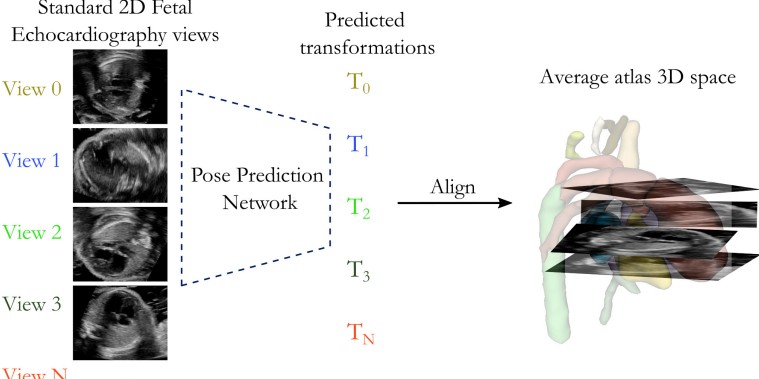

Figure 1: End-diastole frames from standard views of one subject are localised in a standard 3D space via a **F**etal **E**chocardiography **R**egistration **N**etwork. There is no restriction on the number of input slices. Atlas from Uus et al. (2022b).

## 2. Methods

FERN automates the alignment of 2D fetal echocardiography standard views in an average 3D space. Using a Slice-to-Volume Registration Transformer (Xu et al., 2022), it can localise 1-5 standard views in a predefined 3D space (Fig. 1). The network is trained in a supervised manner, simulating 2D slices from real 3D STIC volumes. Random transformations for N slices are sampled from each volume (see Sec. 2.2). FERN predicts transformations aligning 2D slices to 3D space using spatial context across all $N$ slices.

Figure 2 details the transformer architecture (Xu et al., 2022) used for predicting the transformations for each slice.

The main modifications to SVoRT framework are summarised in Table 1. Due to the sparse input, components designed for the full reconstruction are omitted (Xu et al., 2022).

### 2.1. Loss Functions

Predicted transformations are parameterised by three anchor points in a plane (Hou et al., 2018). A plane in 3D may be defined by any three non-collinear points. Here, they are defined as the bottom-left ($P_1$) and bottom-right ($P_2$) corners, and the centre ($P_3$) of the frame. Appendix A includes further details. The network is trained using an L2 loss between predicted ($\hat{P}$) and target ($P$) anchor points (**point loss**). The impact of an **image loss**

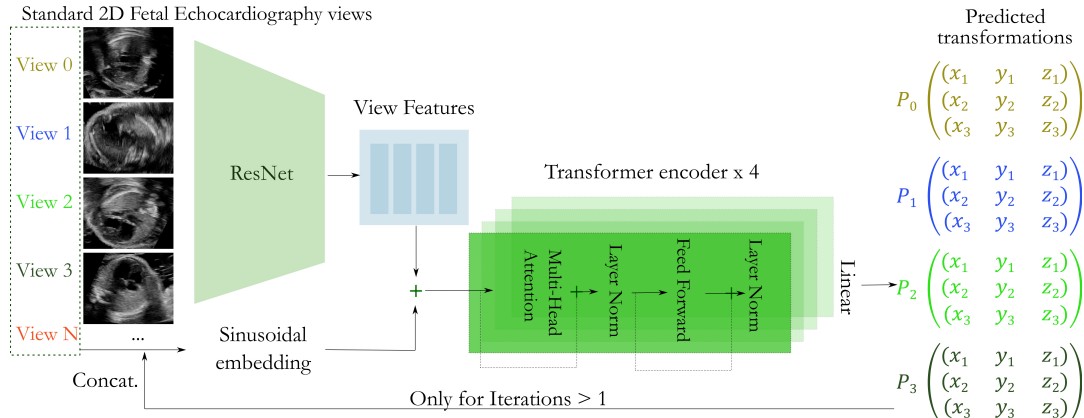

Figure 2: Simplified SVoRT architecture (Xu et al., 2022) used in FERN.

Table 1: Comparison of the Original SVoRT Framework (Xu et al., 2022) and FERN

| Aspect | SVoRT | FERN |
|---|---|---|
| Dataset | Fetal brain MRI | Fetal echocardiography |
| Transformation Sampling | Any orientation | Approx. standard view |
| Slice Transformations in a stack | Correlated | Independent |
| Positional Information | Stack and slice index | View indicator |
| Input Sparsity | Dense, stacks of slices | Sparse, 1-5 views |
| Output | 3D reconstruction | 3D slice transformations |

is also studied via a smooth L1 loss (Appendix B) between the input $y_{in}$ and slices $y_{pred}$, predicted from the original 3D volumes instead of the reconstructions used in the original SVoRT method. The total loss function is

$$\mathcal{L}_{localisation} = \|\hat{P}_1 - P_1\|_2^2 + \|\hat{P}_2 - P_2\|_2^2 + \|\hat{P}_3 - P_3\|_2^2 + \lambda \mathcal{L}_{1smooth}(y_{in}, y_{pred}), \quad (1)$$

## 2.2. Training Transformation Sampling and Positional Embedding

FERN is designed for standard fetal echocardiography views, which are taken at an approximate axial position (see Fig. 3). Therefore, the slice sampling of the 2D training slices from the 3D STIC volumes should reflect this. The coordinate system is centred on the aligned 3D STIC volume origin, with the z-axis aligned axially. Azimuth ($\theta$) and polar ($\phi$) angles capture in-plane and through-plane rotations, respectively. Instead of evenly sampling across a sphere (Hou et al., 2018), the spherical coordinates (azimuth $\theta$ and polar $\phi$ angles) in this approach are:

$$\theta = 2\pi \cdot \text{Uniform}(0, 1), \quad (2)$$

$$\phi \sim \mathcal{N}\left(0, \left(\frac{\pi}{12}\right)^2\right), \quad \text{with } \phi \in \left[-\frac{\pi}{4}, \frac{\pi}{4}\right] \text{ rad.} \quad (3)$$

Here, $\phi$ is sampled from a normal distribution centred at 0 (no through-plane rotation) with $\sigma = 15°$. Angles $> 45°$ are rejected as these represent unrealistic standard views. The full $360°$ in-plane rotations $\theta$ are sampled. See Appendix C for further details.

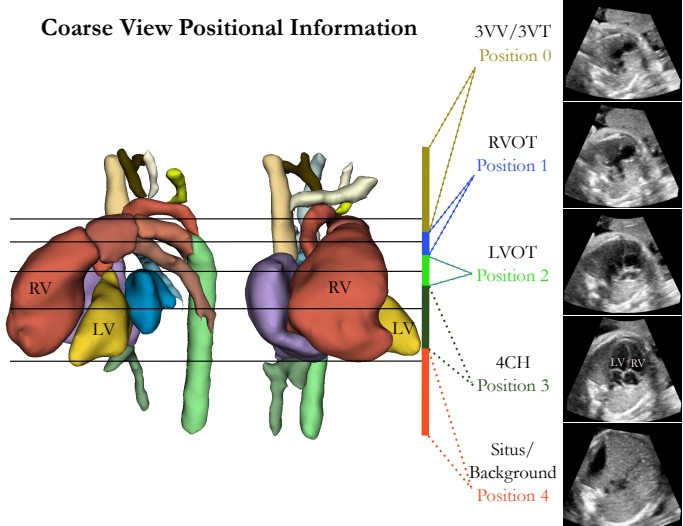

Figure 3: Schematic diagram of the coarse standard view positions: a range of slices are assigned the same position. Atlas from Uus et al. (2022b).

For training, multiple axial slices are sampled from a 3D volume and are independently transformed using in-plane rotations and translations, and through-plane rotations as described above. The initial axial slice position is used to assign a coarse position to each slice, corresponding to a standard view label, as depicted in Fig. 3. The standard view label spans a range of plane positions near the atlas view location, and may be automatically extracted (Baumgartner et al., 2017). To clarify, each axially sampled slice gets assigned a positional label based on proximity to atlas-defined view locations. Proximity is given by the axial slice index before applying 3D transformations (e.g. slices 30-40 get label 3).

## 2.3. Training

The network is trained on dense inputs of up to 94 slices per case, with a 30% chance of randomly switching to 1–5 slices at expected standard view locations, enabling it to learn full spatial context while handling sparse inputs during inference. Data augmentation includes random noise, contrast adjustments, affine transformations, and heart and thorax masking. For this, training slices are randomly multiplied by either a tight heart ROI mask, a dilated heart ROI mask or a mask around the fetal thorax, to ensure robustness to shadows and extracardiac features (see Appendix D).

All experiments are trained for 800000 iterations, using an AdamW optimizer (Loshchilov and Hutter, 2019). Each experiment is repeated five times to improve reliability and account for variability caused by random weight initialisation, stochastic training processes, and data augmentation. Model ensembling is also studied, where the predicted points of all five networks are averaged. The batch size=8, with a linearly decaying learning rate initialised at $2 \times 10^{-4}$. For the image loss, $\lambda = 1000$ (Eqn. 1).

## 2.4. Ablation study

An ablation study is performed to evaluate the impact of SVoRT components. The **Baseline** uses 3 SVT iterations (Xu et al., 2022) with view positional embedding (PE) and image loss (setting $\lambda$ in Eqn. 1 to 0 for *No image loss* or 1000). Components are individually removed: PE, image loss, reduced number of iterations, and the transformer, replaced by a CNN (Yeung et al., 2021).

## 2.5. Training Dataset

The training dataset consists of 3D US (STIC) acquired with a Phillips X6-1 matrix probe. Of an initial 128 volumes from 85 subjects, only 31 volumes from 19 subjects (including 7 for testing) were used due to poor image quality, highlighting the limitations of current 3D acquisitions. The dataset includes 15 Right Aortic Arch (RAA), 8 Coarctation of the Aorta (CoA), and 1 Tetralogy of Fallot case. Despite the small dataset, random slice sampling with through-plane and in-plane rotations and translations greatly increases data variability.

End-diastole 3D STIC frames were rigidly registered to the atlas space (see Appendix E), manually quality-checked, and resampled to $94 \times 94 \times 94$.

## 2.6. Testing Datasets

The testing dataset includes 24 real 2D scans from 7 subjects, each subject having a paired 3D STIC (*Paired real 2D*) and 54 unpaired 2D images from 15 cases (*Unpaired real 2D*), comprising 10 CoA and 12 RAA cases. The 2D paired scans, acquired with a Phillips C9-2 curvilinear probe, include standard views manually aligned to STIC volumes, which were registered to the reference 3D space (Appendix E, F). Paired T2-weighted MRI reconstructions (Uus et al., 2020, 2022a) for these cases were also aligned to their respective 3D STIC volumes. The average gestational age $= 31.17 \pm 1.24$ weeks. An **average atlas** was created from high-quality 3D STIC volumes (Appendix G) and used as an evaluation reference.

## 2.7. Evaluation

Five random in-plane transformations are applied to each 2D image in the *Paired real 2D* dataset, generating 120 test instances (24 slices $\times$ 5 transformations). Each experiment is repeated five times and evaluated using Euclidean Distance on a heart ROI mask ($\text{ED}_{\text{mask}}$), RMSE of translation components ($\text{RMSE}_{\text{trans}}$), and Geodesic Distance (GD, Appendix H). Statistical significance is evaluated using a Wilcoxon signed-rank test against the Baseline. Sec. 3.2 includes a similarity analysis, comparing the real 2D scans to the planes selected from their paired 3D volume (STIC and MRI), given either the manual or predicted transformations, including Multi-Scale Structural Similarity Index (MS-SSIM (Wang et al., 2003)), Normalised Cross-Correlation (NCC) and Normalised Mutual Information (NMI).

For the *Unpaired real 2D* dataset, 2D slices are passed through the network at their acquired orientation, with predicted transformations used to extract corresponding atlas slices, enabling a similarity comparison (Sec. 3.3). Results for the *Paired real 2D* dataset are also included, comparing acquired 2D slices to slices extracted from the atlas using the predicted or manually selected transformation.

An experienced fetal cardiac researcher conducted a blinded qualitative analysis (Sec. 3.4), evaluating predicted and manually selected 3D US (STIC) and atlas planes for feature consistency and presence compared to the 2D input scan, selecting the best plane. For this, expert-annotated anatomical landmarks of high quality STIC were used as references.

## 3. Results

### 3.1. Transformation Accuracy on Paired Real 2D dataset

Table 2: Mean results on manually aligned 2D data. ED = Euclidean Distance, GD = Geodesic Distance. P-values (compared to baseline): *< 0.05, **< 0.005, ***< 0.0005, n.s.=non-significant

| Experiment | $ED_{mask}$ (mm) | $RMSE_{trans}$ (mm) | GD (deg.) | Params. |
|---|---|---|---|---|
| 3 Iters. (Baseline) | $3.264 \pm 1.384$ | $1.4 \pm 0.756$ | $\mathbf{9.749 \pm 4.917}$ | 74M ($\approx 38.12$ h) |
| No PE | $3.748 \pm 1.964$*** | $1.7 \pm 1.024$*** | $10.657 \pm 5.802$* | |
| No Image Loss | $3.548 \pm 1.486$** | $1.576 \pm 0.802$*** | $10.146 \pm 5.309$ n.s. | |
| CNN | $3.996 \pm 1.555$*** | $1.875 \pm 0.921$*** | $10.177 \pm 5.663$ n.s. | 17M ($\approx 15.38$ h) |
| 1 Iter. | $\mathbf{3.198 \pm 1.405}$ n.s. | $\mathbf{1.357 \pm 0.714}$ n.s. | $9.847 \pm 5.345$ n.s. | 32M ($\approx 13.4$ h) |
| 1 Iter. Ensemble | $\mathbf{2.979 \pm 1.269}$ | $\mathbf{1.284 \pm 0.677}$ | $\mathbf{9.074 \pm 4.691}$ | 150M |

Table 2 presents mean error metrics for the *Paired real 2D* dataset. Training with both coarse PE and image loss improves localisation accuracy, while using 3 iterations shows no significant improvement over 1 iteration. However, all average ED and translation errors remain under 4 mm, the maximum inter-subject longitudinal displacement in the manual alignment of 4CH views of the test set. As expected, model ensembling boosts performance significantly, decreasing ED errors by 8.7%, albeit increasing computational cost fivefold. Detailed ensemble results for other ablations are in I, showing similar trends.

The CNN architecture yields higher errors likely due to a lack of positional information, while the transformer, despite fewer parameters, trains ~2 hours faster via task parallelisation. The chosen model is the **ensemble transformer with 1 SVT iteration, PE, and image loss**. Competitive results are also achievable without PE.

### 3.2. Similarity to Paired 3D Acquisition (STIC)

Fig. 4 presents similarity metrics for the *Paired real 2D* dataset, comparing 2D slices to those from the 3D dataset (STIC/MRI) using predicted or manual transformations. Comparable median and mean values are seen across metrics, with no significant difference between manual alignment and GT, indicating competitive network performance.

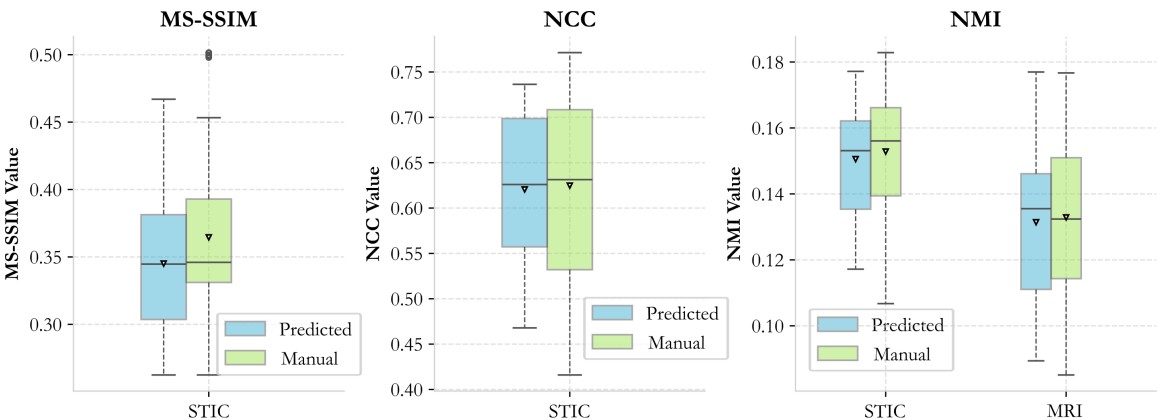

Figure 4: Similarity metrics comparing predicted/manual STIC slices to the 2D.

## 3.3. Similarity to Average Atlas

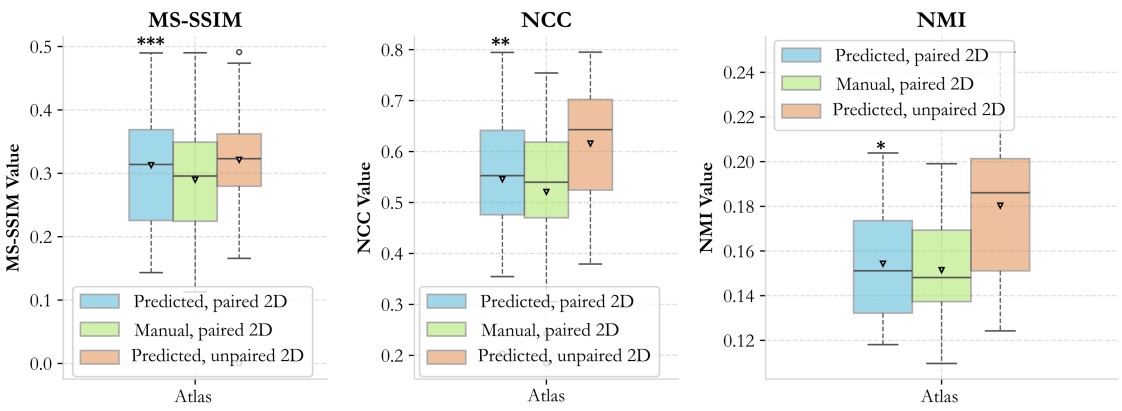

Figure 5: Similarity metrics comparing atlas and 2D US slices.

Fig. 5 compares 2D slices to their corresponding slices in the average atlas, extracted using the predicted transformations. Both *Unpaired real 2D* and *Paired real 2D* datasets are included, with manually aligned results for the latter. Predicted slices show higher mean and median metrics than manually aligned ones (higher=better), with significant differences (tested for paired cases), indicating better alignment to the average 3D space, as seen in Fig. 6, though further validation is needed for clinical landmark correlation.

## 3.4. Qualitative Analysis

The blinded preferred plane selection results are included in Fig. 6, where the closest plane to the 2D is selected. These suggest a similar trend to the similarity metrics reported in Sec. 3.2 and 3.3, where performance between 3D STIC manual or predicted planes is roughly matched, and network alignment to the average atlas space is more accurate than manual alignment. Examples of this are also depicted in Fig. 6 for atlas alignment and 3D STIC.

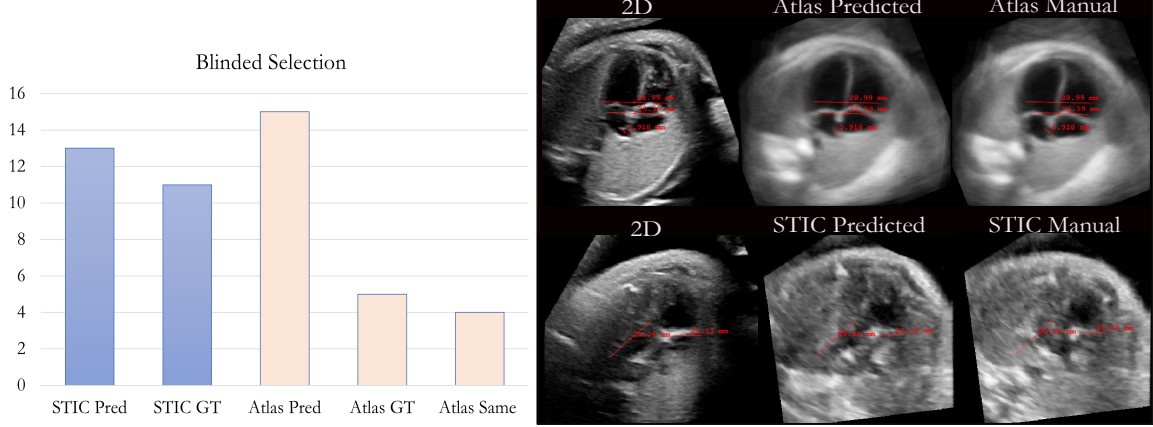

Figure 6: Histogram of the blindly selected preferred slice for the 3D US (STIC) and average atlas based on anatomical similarity to the input 2D slice, with example cases included.

## 4. Discussion

This study proposed a deep learning-based technique for 2D-3D fetal echocardiography alignment, to view high spatio-temporal resolution 2D echocardiography in a lower resolution 3D context. While primarily designed for approximately standard views, in fact, a range of plane angles is captured (see Appendix J.1), and this range could be extended by adapting the orientation sampling during training.

Our motivation for focusing on standard views is their widespread availability, allowing FERN's application within a clinical setting and on retrospective data. By obtaining an accurate alignment into a 3D reference space, deviation from the standard plane can be quantified. The technique could also be used prospectively to provide users with feedback on plane quality. Additionally, a comparison of the standard views with an aligned 3D volume (e.g., an atlas) could provide diagnostic information via case and group-wise comparisons.

The registration network is designed for end-diastole frames. However, given the rapid fetal heart rate, the resulting transformation may be applied to a sequence of frames displaying the whole cardiac cycle within the 3D context. In the future, integrating this method with an automated end-diastole detection tool may allow alignment of approximately axial sweeps with inter-slice motion (see Appendix J.2). Further, modelling direction-dependent artefacts and overcoming cross-modality differences between 2D/3D US may be necessary to build a robust tool suitable for clinical applications, though at present this is alleviated by the use of augmentation during training.

## 5. Conclusion

This study demonstrates the feasibility of aligning sparse standard fetal echocardiography views into an average 3D space using a transformer network, allowing the inclusion of a view positional indicator to improve localisation confidence. FERN enables the combination of high resolution 2D fetal echocardiography with the low resolution 3D acquisitions or atlases to provide 3D context during fetal cardiac examinations.

## Acknowledgments

We would like to acknowledge funding from the EPSRC Centre for Doctoral Training in Smart Medical Imaging (EP/S022104/1).

We thank everyone who was involved in the acquisition and examination of the datasets and all participating mothers. This work was supported by the Rosetrees Trust [A2725], the Wellcome/EPSRC Centre for Medical Engineering at King's College London [WT 203148/Z/16/Z], the Wellcome Trust and EPSRC IEH award [102431] for the iFIND project, the NIHR Clinical Research Facility (CRF) at Guy's and St Thomas' and by the National Institute for Health Research Biomedical Research Centre based at Guy's and St Thomas' NHS Foundation Trust and King's College London. The views expressed are those of the authors and not necessarily those of the NHS, the NIHR or the Department of Health.

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

## Appendix A. Transformation Parametrisation: Anchor Points

The Anchor Points used in this work follow from Hou et al. (2018), which defined a plane using points at the centre, bottom left and bottom right locations of a slice. Parametrising the transformation this way is particularly advantageous for the loss function computation (point loss); which accounts for both rotation and translation parameters, without having to balance out terms from different loss functions in separate losses. Anchor Point parametrisation additionally overcomes challenges present using alternative transformation representations such as Euler Angles, which suffer from non-uniqueness of representation, meaning that the same final rotation may be represented by multiple combinations of different Euler angles. This could lead to optimisation problems and training instability.

To recover the transformation from the Anchor Points, the central Anchor Point $P_2$ defines the $T_z$ translation, i.e. the slice positioning, while the Anchor Points together define the rotation. Given this, the rotation matrix may be computed by finding the x, y, and z-axis normal vectors as

1. X-axis normal vector: $\boldsymbol{v_1} = P_3 - P_1$

2. Z-axis normal vector: $\boldsymbol{n_1} = \boldsymbol{v_1} \times \boldsymbol{v_2}$, where $\boldsymbol{v_2} = P_2 - P_1$

3. Y-axis normal vector: $\boldsymbol{n_2} = \boldsymbol{n_1} \times \boldsymbol{v_1}$

This ensures all axes are orthogonal to each other, to yield the final Rotation matrix by concatenating vectors $\boldsymbol{R} = [\boldsymbol{v1},\, \boldsymbol{v2},\, \boldsymbol{n1}]$ (Hou et al., 2018).

## Appendix B. Smooth L1 Loss

The smooth L1 Loss used as image loss is given by

$$\mathcal{L}_{1smooth}(y_{in}, y_{pred}) = \begin{cases} \frac{0.5(x_n - y_n)^2}{\beta}, & \text{if } |x_n - y_n| < \beta \ (\beta = 0.01) \\ |x_n - y_n| - 0.5 \cdot \beta, & \text{otherwise} \end{cases} \tag{4}$$

where $y_{in}$ are the input slices, and $y_{pred}$ are the predicted slices (simulated from the training STIC volumes). Therefore, instead of using the reconstructed volume and GT volume in the loss function computation, as is done in Xu et al. (2022), this image loss compares each slice at a time, extracted from the volume using either predicted or GT transformations.

## Appendix C. Transformation Sampling

Fig. 7 shows random transformations applied to the unit X, Y and Z-axes vectors (collinear to each axis). Each data point is representative of a plane after applying a random transformation, using the sampling strategy described in 2.2, plotted on a unit sphere.

These are represented as transformed unit vectors. The original vector location with no transformation applied corresponds to the location (1, 0, 0) in the X-axis plot, (0, 1, 0) in the Y-axis plot, and (0, 0, 1) in the Z-axis plot.

The plane surface normals are captured in the Z-axis rotation plot, which displays the through-plane rotations. The X and Y-axis plots also display the in-plane rotation, as each data point represents unit vectors parallel to the X and Y axes.

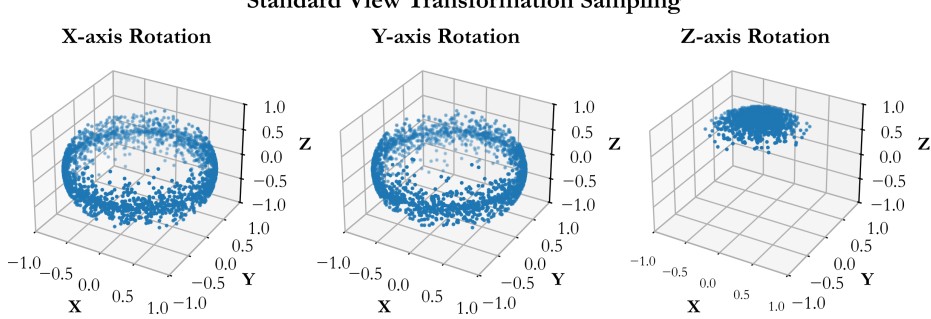

Figure 7: Standard view transformation sampling applied to unit axis vectors (across a unit sphere, covering the range [-1, 1], so unitless), reflecting in-plane rotations (X and Y-axis), and through-plane (Z-axis).

## Appendix D. Random Masking

Fig. 8 illustrates the random masking augmentation strategy. With an equal probability, slices are masked during training using either:

1. No mask

2. Heart ROI mask, Fig. 8, left

3. Dilated heart ROI mask, to encompass further anatomical features

4. Thorax mask, Fig. 8, right.

This masking random augmentation is included as a simple solution to help the network become invariant to extracardiac features, shadows, and the imaging ROI plane. This, coupled with random Gaussian noise, random contrast adjustments and random affine deformations including shearing and scaling, should assist in bridging the gap between the different modalities used for training (3D acquisitions) against the test data (real 2D scans).

## Appendix E. Anatomical Landmarks

The following anatomical landmarks were used for rigid alignment of the 3D STIC volumes used for training:

- Inferior vena cava at the level of the diaphragm

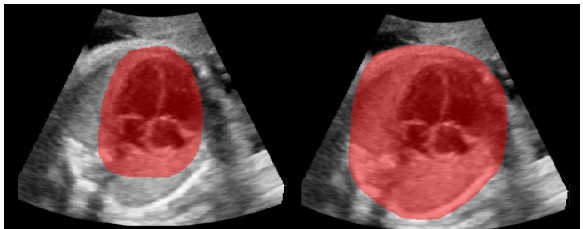

Figure 8: Left: Heart ROI masking, as seen on a 4CH view. Right: Thorax Masking.

- Descending aorta at the level of the diaphragm

- Descending aorta at the level of the left atrium

- Descending aorta at the level of the left atrium

- Left ventricle apex, transverse plane

- Ascending aorta at the level of the right pulmonary artery

- Right ventricular outflow tract (RVOT) just before bifurcation

- Superior vena cava at the level of the 3 vessel view (3VV), right side only

- Crux of the heart

- Carina

- Spine at the level of the three vessel view

## Appendix F. Paired real 2D test set

The *Paired real 2D* dataset consists of 7 unseen cases. Each case consists of a 3D STIC volume, 2D standard view scans and paired MRI volumes (Kuklisova-Murgasova et al., 2012; Uus et al., 2020). The standard view scans were inspected and a high-quality End-Diastole frame was selected.

The corresponding STIC volume slices were examined, and the closest slice to the 2D frame was selected. If the standard view was not visible in the STIC volume, it was excluded, leading to variation in the number of 2D views across the 7 cases (24 slices across 7 subjects).

Corresponding landmarks were recorded for the paired 2D and 3D slices. The 2D slice was then rigidly registered to the matching 3D slice (2D-to-2D registration), allowing to generate a pseudo-volume of 2D slices, by placing the registered 2D slices at the selected 3D slice index location.

Random in-plane transformations can then be applied to the slices in this volume, and a 3D ground truth transformation recovered, used to quantify network performance.

A limitation of this method is the assumption that the selected 2D slice aligns perfectly with an axial slice in the 3D volume, whereas a slight tilt (through-plane rotation) may be present in practice. For this reason, an additional qualitative assessment was performed.

The MRI scans were rigidly aligned to the STIC, which is in the average atlas space, using the same landmarks as described in Appendix E.

## Appendix G. Average Atlas

The average atlas was created using 20 high-quality pre-aligned 3D STIC volumes. The atlas construction steps include:

1. Selection of high-quality initial target image.

2. Rigid, Affine and Free-form deformation of all volumes using MIRTK (Schnabel et al., 2001) to target image. A control point spacing of 6 voxels is used.

3. Spatial average of the registered volumes.

4. Registration bias correction, by averaging all transformations and applying the inverse average transform to the spatial average image.

5. Inspection of results and quality control.

6. Rerun steps 2-5 for 3 iterations, using the corrected average image as the updated target image.

Fig. 9 depicts the resulting atlas across all three dimensions.

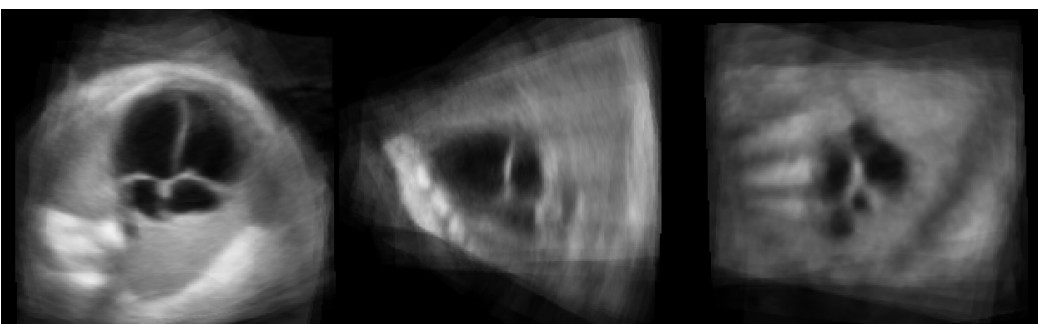

Figure 9: 3D US atlas. The atlas is used for evaluation inside the thorax ROI only (within the fetal ribs).

## Appendix H. Evaluation: Quantitative Metrics

The Euclidean Distance within the masked heart region is calculated as

$$\text{ED}_{\text{mask}} = \frac{1}{N} \sum_{j=1}^{N} \|\hat{P}_j - P_j\|^2,$$

where N are the total number of points within the mask, $\hat{P}_j$ are the predicted points, i.e. the points in the mask after being transformed using the predicted matrix; and $P_j$ are the GT points, obtained using the manually aligned transformation matrix.

The Geodesic Distance is calculated as

$$\text{GD} = \arccos\left(\frac{\text{Tr}(R) - 1}{2}\right)$$

where $R$ is the rotation matrix from the predicted plane to the target, i.e. the composition of the inverse GT rotation matrix with the predicted rotation matrix.

The RMSE of the translation components using

$$\text{RMSE}_{\text{trans}} = \sqrt{\frac{1}{3}\left[(\hat{t}_x - t_x)^2 + (\hat{t}_y - t_y)^2 + (\hat{t}_z - t_z)^2\right]},$$

where $\hat{t}_x, \hat{t}_y, \hat{t}_z$ are the predicted translation components, and $t_x, t_y, t_z$ are the GT translation components for each slice.

## Appendix I. Model Ensembling

A similar performance trend is found compared to non-ensemble, with a decrease in the significance of the results, likely influenced by the reduction in data points after ensembling. The results for 1 Iteration achieving comparable performance to using 3 Iterations remain, despite more than doubling the number of parameters for the latter. Ensembling inference time for a case with four input slices is 0.814 s for 1 Iteration; a fivefold increase compared to a single model pass, at 0.163 s.

Table 3: Mean ± standard deviation test set results on manually aligned 2D data, after ensembling models from five training rounds.

| Experiment | $\text{ED}_{\text{mask}}$ (mm) | $\text{RMSE}_{\text{trans}}$ (mm) | GD (deg.) | Params. |
|:---:|:---:|:---:|:---:|:---:|
| 3 Iters. (Baseline) | **2.951 ± 1.231** | **1.275 ± 0.714** | **8.712 ± 4.028** | 74M ×5 ($\approx$ 38.12 h ×5) |
| No PE | 3.298 ± 1.562 | 1.508 ± 0.785 | 9.574 ± 4.939 | |
| p-value | 0.101 | 0.646 | 0.128 | |
| No Image Loss | 3.118 ± 1.305 | 1.379 ± 0.74* | 9.099 ± 4.363 | |
| p-value | 0.107 | 0.019 | 0.240 | |
| 1 Iter. | 2.979 ± 1.269 | 1.284 ± 0.677 | 9.074 ± 4.691 | 32M ×5 ($\approx$ 13.4 h ×5) |
| p-value | 0.643 | 0.921 | 0.277 | |
| CNN | 3.599 ± 1.245** | 1.666 ± 0.749 ** | 9.145 ± 5.127 | 17M ($\approx$ 15.38 h) |
| p-value | 0.000650 | 0.00124 | 0.966 | |

## Appendix J. Simulated experiments

### J.1. Standard View Variability

Here, the network performance is tested on unseen STIC (3D US) volumes of variable quality. For this, Fig. 10 displays the impact of applying different through-plane angles, which changes the anatomical features.

The network performance remains stable, with an average Euclidean Distance (ED) error of 2.1 mm for no through-plane rotations, 2.1 mm and 2.8 mm after rotating about the X-axis and Y-axis, respectively.

### J.2. Axial Sweep Simulation

Here, a simulated test-time experiment is carried out on 7 unseen 3D volumes of varying image qualities. For this, slices are sampled axially and random inter-slice transformations are applied, as in training. That is, each slice gets randomly rotated (in-plane and through-plane) and translated in-plane up to 30 mm. Slices belonging to a given case are passed through the network, simulating a roughly axial sweep, with significant probe motion.

This is repeated 5 times (i.e. 5 random transformations are applied to each slice) to increase the testing set. Fig. 11 illustrates an example of the aligned sampled slices using the simulated transformations (GT) against the aligned slices using the predicted transformations. There are holes present due to the random rotations applied to each slice, where certain 3D locations may not be sampled (randomly each time). The purpose of this study is to achieve rigid registration, not reconstruction. However, reconstruction methods exist which could be used to reconstruct similar data, filling in the holes (Xu et al., 2023; Kuklisova-Murgasova et al., 2012). This experiment yields a mean Euclidean Distance of $2.3 \pm 1.2$ mm.

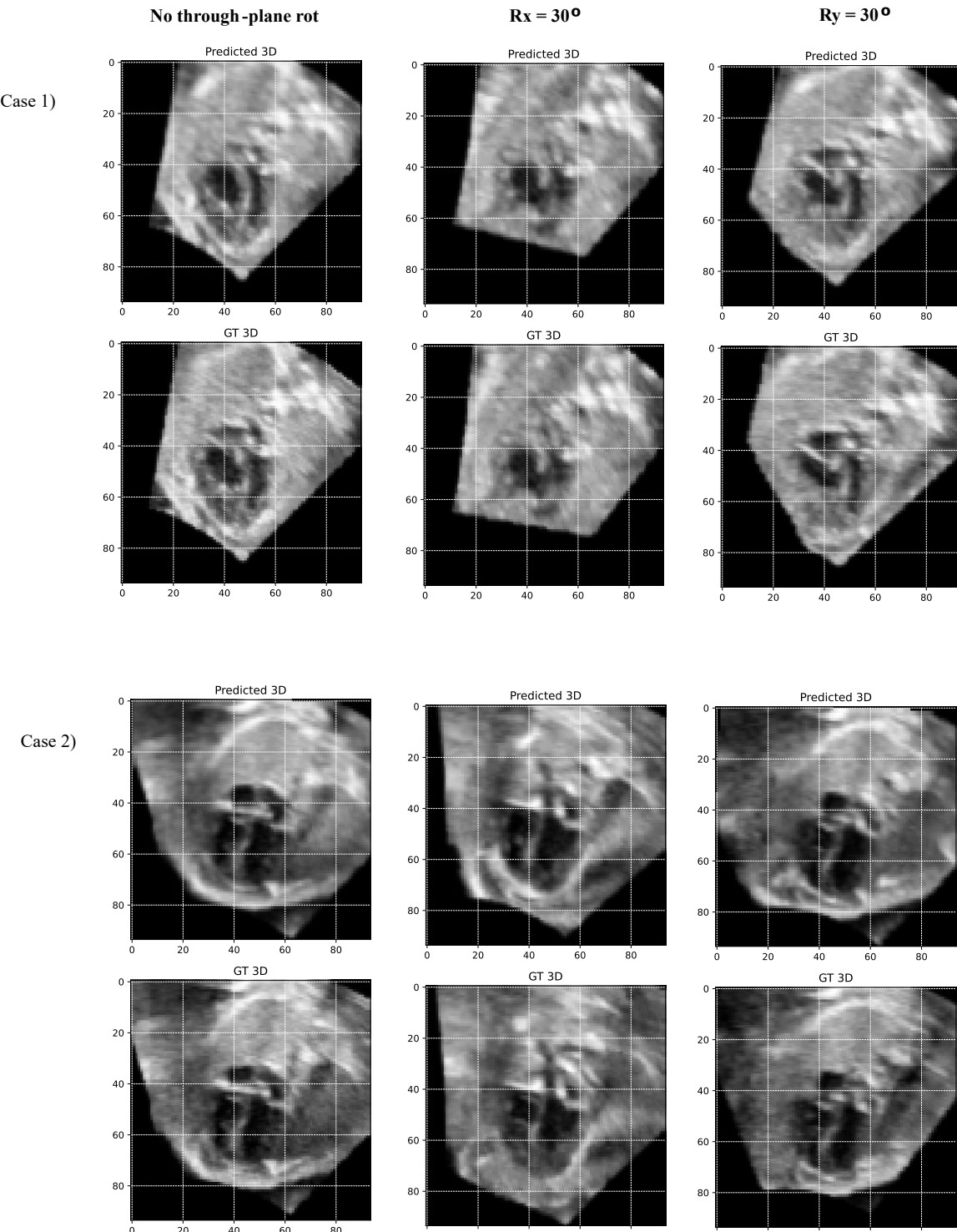

Figure 10: Simulated 4CH and LVOT views from an acquired 3D scan (STIC). Results illustrate the predicted slice (top row) against the GT (bottom row) for two unseen cases. Each column displays the effect of applying a through-plane rotation of 30° about the X-axis (Rx, centre), and the Y-axis (Ry, right).

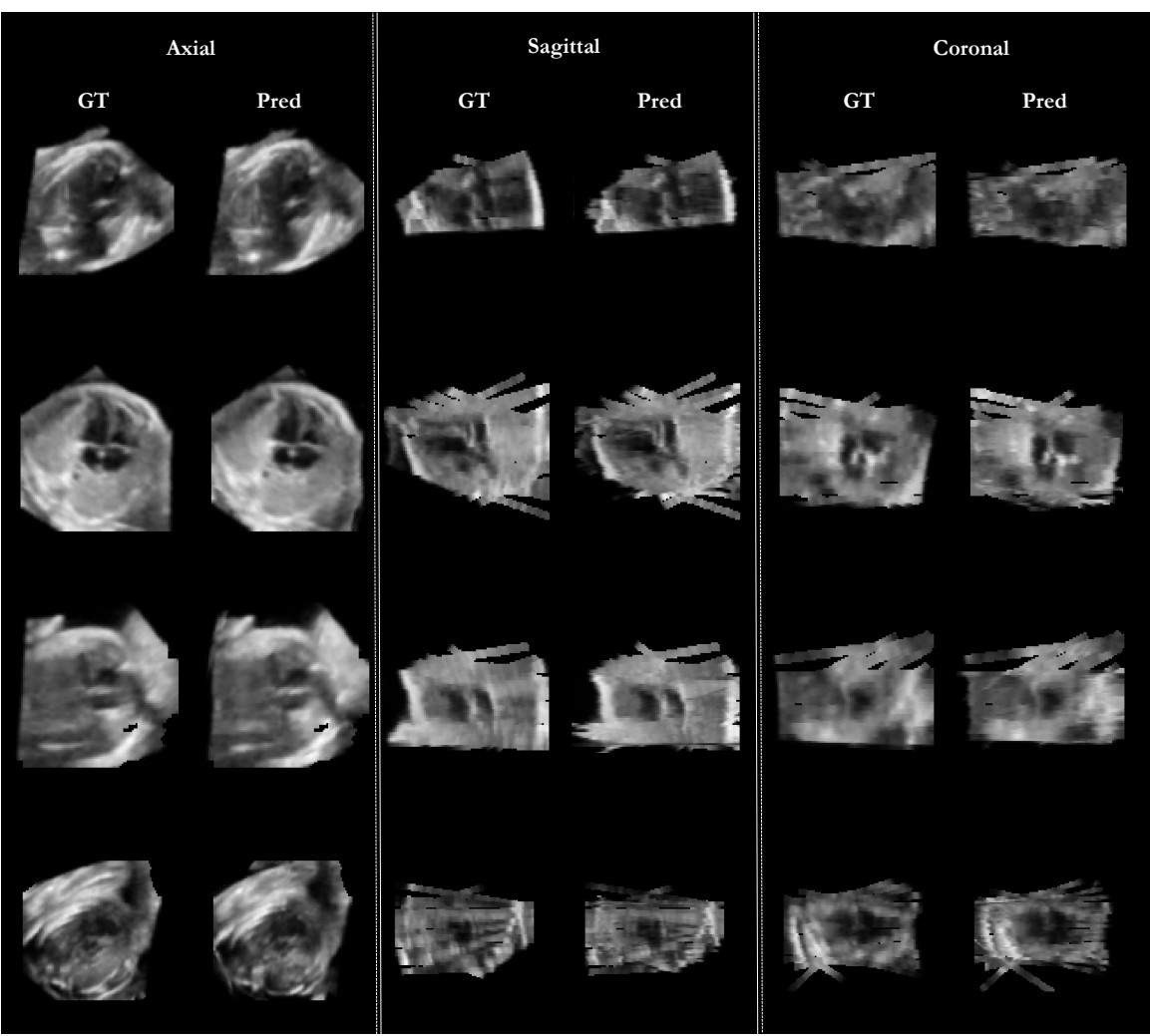

Figure 11: Simulated sweep reconstructions on four unseen cases, displaying the alignment and resulting volumes after registering randomly transformed input slices using the GT transformations (left); and the predicted transformations (right).

