# OpenReview forum: "FERN: A Fetal Echocardiography Registration Network for 2D-to-3D Alignment"
_MIDL.io/2025/Conference — MIDL 2025 Poster_

### Official Review · Reviewer_tuAV · 2025-02-16

**Confidence:** 4
**Preliminary Rating:** 1
**Final Rating:** 4

**Summary:**

This paper describes a method (FERN) for aligning 2D fetal echocardiography images with a pre-determined 3D atlas space. The method builds on a prior method (SVoRT), which uses a transformer network to predict the transformation for an input slice. It is trained using slices sampled from real STIC volumes. The method is evaluated on a real dataset of paired 2D images and STIC volumes.

**Strengths:**

Though similar papers have applied ultrasound slice-to-volume registration to other anatomical locations, to my knowledge, this is the first paper to work with fetal echocardiography. This is a challenging problem and well motivated due to difficulty for clinicians of interpreting this imagery. Another important strength is that the validation is performed on cases with abnormalities, as would be required for any such system to be usable in practice.

**Weaknesses:**

The critical weakness of this paper lies in the exposition of the methods. There are two many unclear and/or missing details for me to understand many important parts of the method, and thus I cannot really comment on its soundness. It seems that a very detailed knowledge of a previous, not particularly well-known paper (SVoRT) is required to begin the understand the method. In such cases, critical details from the cited paper must be included or introduced.

**Detailed Comments:**

- The motivation of the work is to "align [standard] 2D views into an average 3D space ... to provide context to [...] high resolution acquisitions during clinical examinations". However, it is not clear to me how exactly such an alignment would be used by clinicians.
- The authors should consider citing this very relevant paper from MICCAI 2024: https://link.springer.com/chapter/10.1007/978-3-031-72378-0_39
- What precisely do the authors mean by "dense inputs". Without clarification this is not meaningful.
- It appears that all experiments are performed on end-diastole frames. This would seem to be quite a limitation in practice, but the authors do not discuss it as one.
- The authors explain that "predicted transformations are parameterized by three anchor points in a plane" citing Hou et al 2018. Understanding how the transformations are represented as well as the point loss function is critical to understanding the full presented method, and so further detail is required here. What are these anchor points, and how are they defined and chosen?
- Equations 2 and 3 are rather meaningless without first defiining the axes of the coordinate system. Further, I do not understand how phi can be both sampled from a normal distribution and lie within the range -pi/4 to pi/4? Is this done via rejection sampling or clipping to give a truncated normal distribution? What is the motivation for this?
- Figure 2 is poorly explained. What are the units on the X, Y, and Z axes? Is this essentially plotting the position of a certain pixel in the 2D image in 3D space? Personally, I do not think that this figure adds much to the paper, as the behaviour of the sampling can be understand straightforwardly from the equations (except the above comments). I would suggest removing entirely in order to make space for the various missing details from other comments.
- The method of sampling orientations of the simulated 2D frames is described, but how are their positions determined? How does this relate to the definition of standard planes? Is each standard plane a range of axial levels in the atlas space, or a single level?
- Since the method is supposed to perform alignment of 2D slices to a 3D reference space, I do not understand why the model is trained on "dense inputs of up to 94 slices"? Are these parallel from within a 3D volume (which seems quite different from the stated motivation), or neighbouring video frames, which have a known temporal relationship but an unknown spatial relationship in freehand ultrasound.
- The following sentence requires considerable clarification: "Positional embedding uses view type information as prior knowledge". What are the positional embeddings here? Are these the positional embeddings added to each token in the ViT architecture, or something else entirely? What precisely does using "view type information" mean?
- Since ultrasound is direction-dependent (due to shadow artefacts etc), simply sampling images from random orientations does will not provide realistic frames at acquisitions far from the original acquisition direction. The authors should discuss this limitation.
- What are "heart and thorax masking" as augmentations?

**Justification Of The Final Rating:**

The authors have submitted a significantly revised version of the manuscript that goes a long way to clearing up details that were unclear or missing in the initial submission. With the methodology clarified, this is a good paper presenting a good experimental validation of the proposed system. Therefore I am revising my rating to "weak accept".

I do still feel however that, while the authors have clarified this to me in their response, the details of the positional embeddings are still quite unclear in the manuscript itself, which is especially important because they are one of the claimed novel components of the presented system.

**Justification Of The Preliminary Rating:**

This paper cannot be published in its current form because there are too many details about the method (such as position embeddings, why slices are input to the model, details of the architecture used, definitions of the coordinate system used, etc) that are unclear or missing. I have given these in the detailed comments above.

**Questions To Address In The Rebuttal:**

The details of the method from the detailed comments should be clarified.

---

> ### Author Response · Authors · 2025-03-07
>
> We appreciate the reviewer’s time and detailed feedback and have revised our manuscript.
>
> -	it is not clear to me how exactly such an alignment would be used by clinicians.
>
> We have expanded on clinical use in Section 4 and lines 52-56.  Our method is designed for 2D standard views, a globally prevalent acquisition protocol. FERN could be retrospectively used in CHD analysis studies, facilitating downstream tasks by aligning existing acquisitions into a common reference space.  It is clinically valuable to compare different cases within the same 3D context and to an average atlas, as this helps distinguish subject-specific features from pathology and probe/fetal motion, holding educational and diagnostic value.
>
> -	The authors should consider citing this very relevant paper […]
>
> Thank you for this valuable suggestion, now included in lines 71-73
>
> -	What precisely do the authors mean by “dense inputs”.
>
> We have clarified this in lines 66-68 and 91-92. Dense/sparse inputs refers to the requirement for more/less contextual information for a given case in order to generate an accurate transformation prediction, i.e. the number of input slices required.
>
> Attention is computed across input slices (Fig. 2), analogous to the patch-wise attention in a ViT, and used to generate the predicted transformations for all slices for a given case. Thus, more slices per case generally leads to higher accuracy, as there is more spatial context.
>
> -	It appears that all experiments are performed on end-diastole frames. This would seem to be quite a limitation [..]
>
> We added lines 230-234 to discuss the limitation. While the registration network is designed for end-diastole frames, the resulting transformation can be applied to a sequence of frames that contain one or several cardiac cycles. The fetal heart beats quickly (2-3 beats/second), so registration of each individual frame is not essential in practice. Thus the user only needs to select one ED frame to predict the transformation, in order to display the whole cardiac cycle within the 3D context.
>
> -  What are these anchor points, and how are they defined and chosen? […]Equations 2 and 3 are rather meaningless without first defining the axes of the coordinate system. Further, I do not understand how phi can be both sampled from a normal distribution and lie within the range -pi/4 to pi/4[…]
>
> We have added lines 106-108 and Appendix A to explain our Anchor Point definition, and lines 116-118 for coordinate system definition. We apply rejection sampling to exclude angles outside +-pi/4; a threshold empirically determined to exclude unrealistic standard views, line 122.
>
> -	Figure 2 is poorly explained. What are the units on the X, Y, and Z axes?
>
> We have moved this Fig to Appendix C, with explanation in lines 396-404.
>
> -	how are their [simulated 2D slices] positions determined? How does this relate to the definition of standard planes? Is each standard plane a range of axial levels [..]”
>
> We have clarified this in lines 124-129. In short, slices corresponding to different axial levels within an image are assigned a position, covering a range of axial levels in the atlas space (see Fig. 3).
> During training, axial slices, individually perturbed by in-plane rotations and translations; and through-plane rotations, are sampled from a 3D volume. These retain their initial z-translation (even though they are not parallel because we apply through-plane rotations), which is what determines their positional label.
>
> -	why is the model trained on “dense inputs of up to 94 slices”? Are these parallel from within a 3D volume […]
>
> We have added explanation in lines 124-126. We train on slices which densely sample the axial levels (z translation) but with inter-slice motion, including in-plane rotation and translation, through-plane rotations.  We sample more slices during training to allow the network to learn contextual information, as training only on sparse inputs results in insufficient spatial context for network convergence. We randomly switch to an input of 1-5 slices (at any location, line 132), to make the network invariant to the input number of slices.
>
> -	What are the positional embeddings here? [..]What precisely does using "view type information" mean?"
>
> We have rephrased and expanded on this in lines 126-129, and added Fig. 2.
>
> The positional embeddings encode the spatial information regarding the coarse z-translation location of each slice. It is their axial slice position, defined based on closeness to standard views. They are analogous to the positional tokens in the ViT architecture, but instead of having patches and patch embeddings we have slices and slice position embeddings.
>
> -  sampling images from random orientations will not provide realistic frames at acquisitions far from the original acquisition direction.
>
> We have added discussion in lines 234-237
>
> -	What are "heart and thorax masking as augmentations?"
>
> We clarify this in lines 135-137, and Appendix D

---

> > ### Comment · Reviewer_tuAV · 2025-03-07
> >
> > Thanks for the clarifying comments, but I'm not seeing a revised document. Maybe I'm missing something or maybe you forgot to upload it?

---

> > > ### Author Response · Authors · 2025-03-08
> > >
> > > Many thanks, the revised manuscript has been uploaded as part of the supporting material on the Rebuttal (in a zip file).

---

### Official Review · Reviewer_rudr · 2025-02-17

**Confidence:** 5
**Preliminary Rating:** 4
**Recommendation:** Oral
**Final Rating:** 5

**Summary:**

The paper combines and improves upon recent ideas for freehand ultrasound reconstruction. Here the application of Fetal Echocardiography is considered. The method comprises a pose prediction + registration network and removes the need for temporally correlated slices and also dense 3D reconstruction (it hence works on sparse plane samples). The evaluation is thorough and the paper is well written and illustrated.

**Strengths:**

- great clinical motivation and improvements over prior work
- study of different key components of method, e.g. transformer vs CNN, with or w/o positional encoding
- training on simulated data and evaluation on real sweeps with both landmarks and manual inspection

**Weaknesses:**

- I did not find an experiment that demonstrates whether the assumption of correlation of sequentially acquired slices would not bring any further gain
- the inference runtime of the ensembling strategy is not mentioned

**Detailed Comments:**

The dataset is well curated and the comparison to 3D probe data is quite unique to my knowledge. To my understanding the method relies on an average atlas (at least for evaluation) and the visual results in the appendix show a quite blurry 3D image, does this influence the achievable accuracy? While the differences to SVoRT are made fairly clear, I was wondering whether the sparsity of the "reconstruction" does influence the image (compound) loss.

**Justification Of The Final Rating:**

The authors addressed all my remaining concerns and have substantially improved the paper. I would also like to thank them for their detailed rebuttal to all reviewers. I am confident to strongly recommend acceptance now.

**Justification Of The Preliminary Rating:**

I am already quite convinced the paper will make an excellent contribution to the conference and might further increase the score after rebuttal. I would be nice to provide code for reproducibility in the future.

**Questions To Address In The Rebuttal:**

Another interesting baseline would be to only use plane estimation and from that map the relative slice positions using the knowledge of the atlas. Would this be sufficient or is it equivalent to "no image loss"?

**Special Issue:**

Yes

---

> ### Author Response · Authors · 2025-03-07
>
> We would like to thank the reviewer for their time and valuable comments. We will also release our code adaptations and trained network weights. Below, we address the comments.
>
> - I did not find an experiment that demonstrates whether the assumption of correlation of sequentially acquired slices would not bring any further gain
>
> Thank you for raising this important point. We did not include this experiment (as in PlaneInVol) due to the added complexity of our data: the rapid cardiac cycle (110-180 BPM). Real-time sequential acquisition requires first resolving the cardiac phase (e.g., detecting end-diastole frames automatically) before network processing. We have noted this as a limitation in the Discussion (lines 230-234).
>
> We elaborate on this potential in the Appendix J.2, page 18, and Fig. 11 (page 20) where we simulate an axial sweep (end-diastole) from 3D volumes with random slice motion, achieving a low mean error of 2.3 mm. This shows that more context improves accuracy, though real sparse data yields a comparable 3.2 mm error (Table 2); however this may also be due to cross-modality differences.
>
> - the inference runtime of the ensembling strategy is not mentioned
>
> We have amended our manuscript to include this in Appendix I, lines 479-481.
>
> - The dataset is well curated and the comparison to 3D probe data is quite unique to my knowledge. To my understanding the method relies on an average atlas (at least for evaluation) and the visual results in the appendix show a quite blurry 3D image, does this influence the achievable accuracy?
>
> The atlas is used only for evaluation, where we calculate similarity of the real 2D data with the selected atlas plane. The similarity is only calculated inside the cardiac ROI, which is relatively well defined.
>
> We have clarified this in the manuscript, Caption of Fig. 9 (page 16). We also conduct analysis on 3D STIC acquisitions to have more complete picture of method performance.
>
> - While the differences to SVoRT are made fairly clear, I was wondering whether the sparsity of the "reconstruction" does influence the image (compound) loss.
>
> We have had to adapt the image loss to our specific case, as we randomly switch to sparse inputs during training. For this, instead of using a loss function comparing a full reconstruction, we compare the acquired slice (from the GT volume) given the GT transformations against the acquired slice given the predicted transformations. We have clarified this in line 111; and 391-394 in Appendix B.
>
> - Another interesting baseline would be to only use plane estimation and from that map the relative slice positions using the knowledge of the atlas. Would this be sufficient or is it equivalent to "no image loss"?
>
> This is an interesting point, however the standard plane location does not occupy exactly the same position in the reference space for different subjects. This is because the standard planes are defined by the visibility of specific anatomical landmarks, and due to inter-subject anatomical variability and pathology, this plane can vary from case to case. For this reason, we manually aligned 2D slices to their corresponding location in a 3D acquisition of the same subject, explained in Appendix F. Subsequently 3D images were aligned to the atlas space using anatomical landmarks selected by experienced fetal cardiologist. We found that position in the atlas could vary by as much as 4 mm in z direction only, demonstrating, that standard planes in the atlas are not necessarily well defined.
>
>  The experiment where we use “No image loss” refers to training using just Anchor Points, which we have clarified in line 147. Appendix A expands on the Anchor point loss.

---

### Official Review · Reviewer_tNeu · 2025-02-21

**Confidence:** 4
**Preliminary Rating:** 2
**Final Rating:** 2

**Summary:**

The paper presents a method for registration of standard view 2D US images to an average 3D US volume, using positional indicators (which are the standard view indications) and a transformer network to predict the affine transformations. The registration was evaluated with image similarity metrics to the average atlas and qualitative scoring compared with manual position estimates.

**Strengths:**

The method is examined using ablation experiments that are described in detail.

The work includes a structured qualitative evaluation to assess the method performance, based on clinically relevant landmarks.

**Weaknesses:**

The method operates only on standard US planes. Acquiring standard US planes requires significant expertise: someone who can do this, will know the context of the acquired plane. As the method does not operate on non-standard planes, the value of the method is unclear. It does not allow less experienced users to acquire a high quality plane and learn about the context by using the registration tool, and it cannot be used for extracting standard planes or constructing volumes from ultrasound sweeps for analysis, as used in some cases where no experienced US operators are available.

The authors indicate the lower temporal resolution of STIC is problematic, however the standard 2D planes that are aligned with the 3D volume are from one specific timepoint (end-diastole) and thus do not improve temporal resolution.

Selecting 2D slices from a 3D volume does not result in realistic 2D US images. It isn’t mentioned whether the authors considered this or how they corrected for this.

**Detailed Comments:**

The numbering of the figures is not consistent with the order in which they are referenced.

The manuscript is inconsistent on whether 1-5 or 1-6 slices are used as input.

The authors state in section 3.3 that FERN’s performance is superior when comparing similarity metrics of the 2D registered image with the 3D volume to the manual registration. However, FERN is optimized for minimizing these dissimilarity metrics, and it is unclear how well these correlate with relevant clinical landmarks, so this conclusion seems too strong.

**Justification Of The Final Rating:**

While the authors have done an admirable job of revising the manuscript and addressing concerns of the reviewers, the justification of the added benefit of the method has not convinced me. The two stated contributions of the method are "(1) the use of a view positional indicator" and "(2) the ability to handle sparse inputs consisting of only 1-5 standard view slices in any orientation.". The first is only relevant if a skilled operator is available, but the problem being solved is a subset of the skills required to acquire these views in the first place. The second contribution is irrelevant in 2D ultrasound exams; all 2D ultrasound systems output temporally dense video streams, and in 2D B-mode imaging, there is no benefit to artificially lowering the frame-rate. No additional justification of given for the necessity of such sparse inputs, and there is no clear link between this stated contribution and the clinical problem under consideration.

The authors argue that the method could be extended to support extracting standard views from ultrasound sweeps for situations where a trained operator is not available, but the main difficulty in positioning such sweeps lies in dealing with the large amounts of acoustic shadows crossing through the slices and resulting in partially obscured structures. The sweep experiments added to the appendix use data sampled directly from a 3D volume, where such shadows are not present at all. While the authors use partial masking to claim robustness to shadows, the images are never masked with shadows crossing the to-be-aligned structures (which is the main difficulty of aligning views that are not carefully selected shadow-free standard planes), so the presented experiments are highly unrepresentative of the suggested clinical use. The same is true for assessing view quality and applying the method for operator guidance; while this would be a useful application of a similar method, neither of the stated contributions are a valuable step in this direction.

**Justification Of The Preliminary Rating:**

The requirement of acquiring standard view planes makes the intended application of the presented method unclear: someone who can do this, will by necessity already know the context of the acquired plane.

**Questions To Address In The Rebuttal:**

Currently, the added value of the method is unclear. The authors should include a discussion on their envisioned application of the presented method.

What is the clinical experience of the rater in the qualitative experiments?

---

> ### Author Response · Authors · 2025-03-07
>
> We thank the reviewer for the insightful feedback, below we address the comments.
>
> -  Currently, the added value of the method is unclear. The authors should include a discussion on their envisioned application of the presented method.
>
>  and have amended the manuscript to include details to clarify our study’s motivation, limitations, and future applications:
>
> -- Introduction, lines 52-56
> -- Discussion, lines 224-229
>
> We would like to highlight that the transformations sampled during training, although restricted, capture a range of slice orientations. The technique is therefore applicable to non-standard views that do not significantly deviate from high-quality standard views. We have clarified this in lines 221-223, Appendix J.1, page 17, with examples in Fig. 10.
>
> While our current tool would not work for planes that significantly deviate from standard views (for instance, sagittal planes), this can be easily addressed by sampling a wider range of transformations. We have discussed this in lines 221-223.
>
> Additionally, the real 2D data available to us for testing consists only of 2D standard views. These acquisitions are prevalent worldwide, representing the simplest starting point for testing registration accuracy on real fetal echocardiography data. FERN could be used retrospectively across centers,  facilitating downstream tasks in CHD studies while enhancing data interpretation by comparing different cases to a 3D reference. Specifically, allowing to discern in a controlled and well defined acquisition protocol: (a) subject-specific anatomy (b) disease-specific features (c) probe and fetal motion. This spatial information could have diagnostic and educational utility.
> We expect future work to build towards a full reconstruction strategy, targeting more view locations.  Cardiac cycle detection would allow to use our tool to align roughly axial sweeps (there is no limit to the number of input scans), even with inter-slice motion. See Appendix J.2, pages 18 and 20 (Fig. 11).
>
> - The authors indicate the lower temporal resolution of STIC is problematic, however the standard 2D planes that are aligned with the 3D volume are from one specific timepoint (end-diastole) and thus do not improve temporal resolution.
>
> We clarify this in the discussion, lines 230-234. While the registration network is designed for end-diastole frames, the resulting transformation can be applied to a sequence of frames that contain one or several cardiac cycles. The fetal heart beats quickly (2-3 beats per second), so registration of each individual frame is not essential in practice. Thus, the user only needs to select one ED frame to predict the transformation, in order to display the whole cardiac cycle within the 3D context.
>
> - Selecting 2D slices from a 3D volume does not result in realistic 2D US images. It isn’t mentioned whether the authors considered this or how they corrected for this.
>
> This is indeed a challenging problem, which we mitigate by augmenting our training data, as described in lines 133-137 with: Random masking, Gaussian noise, contrast adjustments and affine deformations. Masking makes the network invariant to extracardiac features such as shadows and US imaged area (the trapezoid ROI). An example of the masked areas is included in the Appendix E, Fig. 8. We randomly mask the input, switching between four options: No masking; Masking around the cardiac ROI; Dilating this mask to capture a larger ROI; Thorax masking.
>
> While we have demonstrated that training on simulated data with sufficient augmentation produces accurate results on real 2D data; future work may seek to expand on these cross-modality differences, which may become more challenging as this expands towards predicting any pose. We have added this to our future work discussion, lines  234-237.
>
> - The numbering of the figures is not consistent with the order in which they are referenced. The manuscript is inconsistent on whether 1-5 or 1-6 slices are used as input.
>
> We thank the reviewer for raising these points and have corrected them.
>
> -	The authors state in section 3.3 that FERN’s performance is superior when comparing similarity metrics of the 2D registered image with the 3D volume to the manual registration. However, FERN is optimized for minimizing these dissimilarity metrics, and it is unclear how well these correlate with relevant clinical landmarks, so this conclusion seems too strong.
>
> We have amended this in lines 210-211. The average atlas is however only used for evaluation.
>
> - What is the clinical experience of the rater in the qualitative experiments?
>
> We have added this to lines 183 and 186. An experienced fetal cardiac researcher conducted the blinded qualitative analysis, consulting expert cardiologists for unclear cases. Expert-annotated anatomical landmarks (by fetal cardiologists, Appendix E) from high-quality STIC scans, all aligned to the common 3D reference, provided reliable benchmarks.

---

> ### Author Response · Authors · 2025-03-14
>
> We appreciate the reviewer’s feedback and concerns and thank them again for their time. To clarify our motivation, our method registers high-resolution 2D views into a 3D reference space to enhance anatomical analysis, case comparisons, and atlas-based assessments. Unlike current 3D acquisitions, which suffer from lower resolution and motion artifacts, our approach preserves the superior quality of 2D fetal echocardiography—even exceeding fetal cardiac MRI reconstructions in terms of spatial resolution. The primary goal is to integrate high-quality 2D+t scans into a 3D context to aid in visualizing congenital heart disease.
>
> Regarding sparsity, we are not lowering the frame rate but using standard fetal screening data, an acquisition protocol widely available retrospectively, which already relies on key diagnostic views rather than dense sampling. Therefore, our motivation is to develop a tool that works on prevalent acquisition protocols (with any frame rate, as typically each view is assessed over a temporal window). Placing such views in a 3D context improves case comparisons (for example, cases with similar diagnostic concerns), vessel course assessment (for instance, in aortic arch anomalies), and correlation with other 3D modalities (e.g., STIC), making subtle features more discernible.
>
> Our method provides an alternative for anatomical assessment without requiring full volumetric scans. However, we acknowledge that it currently requires expertise comparable to that of screening sonographers, where it may aid them in assessing case-specific cardiac anatomy.

---

### Author Rebuttal · Authors · 2025-03-07

**Rebuttal:**

The authors would like to thank the reviewers for detailed and supporting feedback and comments, which thoroughly address the weaknesses of the paper. The main updates to the papers are:

-	Addition of a Discussion section, where we detail our proposed clinical use-case of FERN, limitations and avenues for future work
-	Addition of architectural details, and clarification regarding transformation parametrisation, training slice sampling, positional embedding and training augmentation
-	Addition of Appendix sections with simulated test-time experiments, where we discuss applicability to non-standard views and axial sweeps, testing the gain further spatial context brings

**Supporting Material:**

/attachment/1fe8675c52ded3ac2f9153b02f3bd1731f23504e.zip

---

### Meta-Review · Area_Chair_EY7p · 2025-03-20

**Recommendation:** Accept (Poster)
**Confidence:** 4

**Metareview:**

The paper outlines a method for aligning 2D ultrasound views of congenital heart disease with a standard 3D view. This presents a significant challenge, prompting the authors to adapt an existing algorithm called SVoRT. Overall reviews were mixed, primarily because of a lack of detail in the original submission. The authors effectively addressed the reviewers' comments in a thoroughly revised version. The clinical motivation behind the work is evident. The goal is not to produce a 3D reconstruction, but to enhance the repeatability of imaging by positioning slices within a standardized 3D coordinate system. Authors provide the code of their method. I believe that this work will be of interest to the MIDL community and recommend acceptance.

In a camera-ready version, authors should address:
- Reviewer tuAV: *The details of the positional embeddings are still quite unclear in the manuscript itself and should be further clarified in the manuscript.*
- Table 2 should be formatted to meet the page width.